# Gfi1 Loss Protects against Two Models of Induced Diabetes

**DOI:** 10.3390/cells10112805

**Published:** 2021-10-20

**Authors:** Tiziana Napolitano, Fabio Avolio, Serena Silvano, Sara Forcisi, Anja Pfeifer, Andhira Vieira, Sergi Navarro-Sanz, Marika Elsa Friano, Chaïma Ayachi, Anna Garrido-Utrilla, Josipa Atlija, Biljana Hadzic, Jérôme Becam, Anette Sousa-De-Veiga, Magali Dodille Plaisant, Shruti Balaji, Didier F. Pisani, Magali Mondin, Philippe Schmitt-Kopplin, Ez-Zoubir Amri, Patrick Collombat

**Affiliations:** 1Faculté des Sciences, Université Côte d’Azur, CNRS, Inserm, iBV, Parc Valrose, 06108 Nice, France; Tiziana.NAPOLITANO@univ-cotedazur.fr (T.N.); Serena.SILVANO@univ-cotedazur.fr (S.S.); anja.d.pfeifer@gmail.com (A.P.); andhira.vieira@gmail.com (A.V.); marikafriano@gmail.com (M.E.F.); Chaima.AYACHI@univ-cotedazur.fr (C.A.); garridoanna92@gmail.com (A.G.-U.); Jerome.BECAM@univ-cotedazur.fr (J.B.); Anette.SOUSA-DE-VEIGA@univ-cotedazur.fr (A.S.-D.-V.); Magali.PLAISANT@univ-cotedazur.fr (M.D.P.); Ez-Zoubir.AMRI@univ-cotedazur.fr (E.-Z.A.); 2Department of Biochemistry and Molecular Biology, University of Southern Denmark, 5230 Odense, Denmark; fabioavolio@bmb.sdu.dk; 3Research Unit Analytical BioGeoChemistry, Helmholtz Zentrum München, German Research Center for Environment Health, 85764 Neuherberg, Germany; sara.forcisi@helmholtz-muenchen.de (S.F.); schmitt-kopplin@helmholtz-muenchen.de (P.S.-K.); 4German Center for Diabetes Research (DZD), 85764 Neuherberg, Germany; 5CIRAD, UMR AGAP, 34398 Montpellier, France; sergi.navarro_sanz@cirad.fr; 6TAAM, CNRS, UPS 44, 45100 Orleans, France; josipa.atlija@cnrs-orleans.fr; 7Pediatric Oncology & Hematology Department, Centre Hospitalier Universitaire de Nice, Hopital Archet 2, 06202 Nice, France; hadzic.b@chu-nice.fr; 8PlantaCorp GmbH, 20097 Hamburg, Germany; Shruti.balaji@plantacorp.com; 9Medicine Faculty, Université Côte d’Azur, CNRS, LP2M, 06003 Nice, France; Didier.PISANI@univ-cotedazur.fr; 10Pôle Imagerie Photonique, Bordeaux Imaging Center, Université de Bordeaux, UMS 3420 CNRS-US4 Inserm, 33076 Bordeaux, France; magali.mondin@u-bordeaux.fr

**Keywords:** pancreas development, amylase, ghrelin, mouse model, high-fat diet

## Abstract

**Background:** Although several approaches have revealed much about individual factors that regulate pancreatic development, we have yet to fully understand their complicated interplay during pancreas morphogenesis. Gfi1 is transcription factor specifically expressed in pancreatic acinar cells, whose role in pancreas cells fate identity and specification is still elusive. **Methods:** In order to gain further insight into the function of this factor in the pancreas, we generated animals deficient for *Gfi1* specifically in the pancreas. *Gfi1* conditional knockout animals were phenotypically characterized by immunohistochemistry, RT-qPCR, and RNA scope. To assess the role of Gfi1 in the pathogenesis of diabetes, we challenged *Gfi1*-deficient mice with two models of induced hyperglycemia: long-term high-fat/high-sugar feeding and streptozotocin injections. **Results:** Interestingly, mutant mice did not show any obvious deleterious phenotype. However, in depth analyses demonstrated a significant decrease in pancreatic *amylase* expression, leading to a diminution in intestinal carbohydrates processing and thus glucose absorption. In fact, *Gfi1*-deficient mice were found resistant to diet-induced hyperglycemia, appearing normoglycemic even after long-term high-fat/high-sugar diet. Another feature observed in mutant acinar cells was the misexpression of ghrelin, a hormone previously suggested to exhibit anti-apoptotic effects on β-cells in vitro. Impressively, *Gfi1* mutant mice were found to be resistant to the cytotoxic and diabetogenic effects of high-dose streptozotocin administrations, displaying a negligible loss of β-cells and an imperturbable normoglycemia. **Conclusions:** Together, these results demonstrate that Gfi1 could turn to be extremely valuable for the development of new therapies and could thus open new research avenues in the context of diabetes research.

## 1. Introduction

The pancreas plays a pivotal role in regulating glucose homeostasis by controlling both carbohydrates absorption and peripheral glucose handling. This gland orchestrates endocrine and exocrine responses due to the anatomical and functional interplay of its three main components: islets of Langerhans, acini, and ducts. Acinar cells are specialized for the synthesis of digestive enzymes: proteases, lipases, amylases and nucleases, degrading dietary proteins, lipids, carbohydrates, and nucleic acids [1]. These are transported and emptied into the duodenum by ductal cells, forming a ductal network. Pancreatic endocrine cells, in turn, are organized into highly vascularized clusters termed islets of Langerhans. These cells are classified into β-, α-, δ-, PP-, and ε-cells, secreting insulin, glucagon, somatostatin, pancreatic polypeptide, and ghrelin, respectively [2].

The main role of the endocrine pancreas is to ensure a homeostatic control of the glycaemia by secreting insulin (acting to decrease glycemia upon need thereof) and glucagon (whose role is to increase glycemia). In addition, somatostatin and pancreatic polypeptide regulate the secretion of numerous hormones and enzymes [3]. Abundant during pancreatic embryonic development, ghrelin-expressing cells represent less than 1% of endocrine cells in adult mice. The role of ghrelin in the pancreas has been poorly elucidated; however, a paracrine/autocrine role in the regulation of β-cell survival has been proposed [4,5,6].

Type 1 diabetes mellitus is a chronic autoimmune disease characterized by a severe deficiency of insulin secretion due to pancreatic β-cells loss [7]. Conversely, type 2 diabetes mellitus is associated with a profound metabolic dysregulation, resulting from impaired insulin secretion, insulin resistance, or a combination of both [8].

Genome-wide association studies highlighted several variants associated with high risks of developing diabetes; these matching genes are crucially involved in pancreatic cell differentiation [9,10]. Therefore, further understanding the molecular mechanisms regulating pancreas development remains one of the best approaches to potentially discover new therapeutic targets for diabetes.

During mouse embryonic development, the pancreatic domain along the primitive foregut is defined at embryonic day 8.5 (E8.5) by the expression of *Pdx1* within multipotent pancreatic progenitor cells [11,12]. Subsequently, the pancreatic epithelium thickens, forming a highly branched tubular structure and compartmentalizes into tip and trunk domains. The trunk predominantly gives rise to endocrine and ductal cells, whereas the tip cells are rapidly restricted to an acinar cell fate [13,14]. The commitment of a given MPC to an acinar or endocrine/ductal fate is determined by the mutually antagonistic action of Nk6 *Homeobox* (Nkx6) factors and pancreas-associated transcription factor 1a (Ptf1a). Hence, starting from E12.5, *Ptf1a* becomes progressively restricted to peripheral tips, while *Nkx6.1* and *Nkx6.2* are progressively confined to trunks [15].

In silico genomic studies suggested that growth factor-independent 1 (*Gfi1*) might regulate early pancreatic development machinery as a downstream target of Ptf1a [16]. *Gfi1* gene encodes a 55 kDa transcriptional repressor containing six C-terminal zinc finger domains (DNA-binding) and an N-terminal-located SNAG (SNAIL/Gfi-1) transcriptional-repressor domain [17]. Describing a *Gfi1* full knock-out mouse line, Qu et al. reported that, in the absence of *Gfi1*, the pancreas fails to establish a centroacinar cell population [18]. Lacking the interface between the acinar cells and the ductal tree, zymogen granules accumulate in the acinar tissue, leading to tissue damage.

To fully dissect the role of Gfi1 in pancreatic development, we generated a conditional knockout model, inactivating *Gfi1* solely in the pancreas. The phenotypical characterization of this mouse line allowed us to demonstrate that *Gfi1* is required for pancreatic acinar cell full development. Specifically, we showed that *Gfi1*-deficient acinar cells do not exit the cell cycle and express diminished levels of amylase. Importantly, such a decrease in the enzyme responsible for the processing of carbohydrates readily assimilable by the intestine appeared beneficial. Indeed, *Gfi1*-deficient mice were found able to maintain a normal glycemia despite the long-term administration of a high-fat diet. As important was the finding that animals lacking *Gfi1* in their pancreata were protected from streptozotocin-mediated diabetes, most likely due to the misexpression of Ghrelin in acinar cells.

## 2. Materials and Methods

### 2.1. Generation of Gf1Cre::RosaLac and Pdx1Cre::Gfi1cko Mouse Lines

Gfi1Cre::RosaLac mice were obtained by crossing Gfi-Cre mice [19] (expressing a *Cre recombinase* under a *Gfi1 promoter*) with the RosaLac reporter line [20] (harboring a transgene composed of the ubiquitous *ROSA26 promoter* upstream of a loxP-STOP-LoxP cassette followed by the *β-galactosidase cDNA)*.

Pdx1Cre::Gfi1cko animals were generated by crossing Pdx1Cre animals [21] (harboring a transgene composed of the *Pdx1* promoter driving the expression of the *Cre recombinase* in all multipotent pancreatic progenitor cells) with Gfi1^fl/fl^ animals [22] (in which the 4th and 5th exon of the Gfi1 gene are flanked by two LoxP sites). In the resulting bitransgenic mice, Cre-recombinase mediates the excision of the genomic region encoding *Gfi1 DNA binding domain*. Gfi1^fl/fl^ littermates were used as controls.

### 2.2. Animal Manipulations

Mouse colonies were maintained following European ethical regulations. At weaning, mice were fed with either a standard diet (A03, SAFE, Rosenberg, Germany) or HFCD (260HF, SAFE, Rosenberg, Germany). Glycemia was measured with ONETOUCH glucometer (LifeScan, Malvern, PA, USA, Table 1), following 5 h of starvation. Food intake was measured as the difference in weight between the food provided to a cage and that remaining after 24 h. Cellular proliferation was assessed by providing mice with 1 mg/mL of 5-bromo-2′-deoxyuridine, BrdU, dissolved in drinking water. Three-month-old mice were starved for 5 h prior streptozotocin (STZ) treatment. STZ was dissolved in 0.1 M of sodium citrate buffer (pH 4.5) and administered intraperitoneally at a dose of 135 mg/kg.

### 2.3. Gene Expression Analyses

For RNA extraction, a small piece of pancreas or 200 islets of Langerhans were incubated in cold RNA later solution (Table 1) at 4 °C. When RNA extraction from islets of Langerhans was required, 1 mg/mL of collagenase (Table 1) diluted in MEM medium (Sigma Aldrich, St. Louis, MO, USA) was injected into the main pancreatic duct, and pancreatic tissue was incubated at 37 °C for 10 min. Subsequently, islets were separated by crude tissue fractionation through a Histopaque (Sigma Aldrich, St. Louis, MO, USA, Table 1) gradient. The following day, islets were handpicked and pelleted. Total RNA was therefore extracted using an Rneasy Mini Kit (Qiagen, Germantown, MD, USA, Table 1), according to the manufacturer’s instructions. RNA concentration and integrity were determined using Agilent 2100 Bioanalyzer system (Agilent Technologies, Santa Clara, CA, USA). cDNA synthesis was performed using Superscript choice system (Table 1). Quantitative RT-qPCR was carried out using the QuantiTect SYBR Green RT-PCR Kit (ThermoFisher, Waltham, MA, USA, Table 1) and validated primers (Qiagen, Germantown, MD, USA), using GAPDH as internal control for normalization purposes.

### 2.4. Immunohistochemistry

For immunohistochemical analyses, pancreatic tissue was isolated, fixed, sectioned, and assayed, as described previously [23] using DAPI as counterstain. Pictures were processed using ZEISS Axioimager Z1 (Carl Zeiss AG, Oberkochen, Germany) equipped with the appropriate filter sets and a monochrome camera, with Axiovision software (Axiovision Rel. 4.8, New York, NY, USA) from ZEISS.

### 2.5. ELISA Immunoassay

Twenty mg of fresh pancreatic tissue were homogenized in non-denaturing lysis buffer (20 mM of Tris-HCl pH = 8, 137 mM of NaCl, 1% NP40, 2 mM of EDTAm and 0.1 mg/mL of PMSF). The total protein concentration was assessed using a BCA protein assay (ThermoFisher, Waltham, MA, USA, Table 1), and 100 μg of protein lysate were used for the ELISA assay (Bertin Technologies, Montigny-le-Bretonneux, France, Table 1), following manufacturer’s instructions.

### 2.6. X-Gal Staining

Pancreatic tissue was fixed for 30 min at 4 °C (1% paraformaldehyde; 0.2% glutaraldehyde; 0.02% NP40). Subsequently, each sample was incubated overnight at 4 °C in a 25% sucrose solution and embedded in tissue-freezing medium (Table 1). Sixteen-μm sections were incubated with an X-gal staining solution (10 mg/mL of 5-bromo-4-chloro-3-indolyl-β-d-galactoside, 5 mM of K_3_Fe(CN)_6_, 5 mM of K_4_Fe(CN)_6_, and 2 mM of MgCl_2_, PBS) at 37 °C overnight.

### 2.7. RNAscope

Pancreatic tissue was fixed in formalin for 16 h at 25 °C. Subsequently, samples were dehydrated, embedded in paraffin, and sectioned into 6-μm slides. In situ RNA detection was performed using an RNAscope kit (Advanced Cell Diagnostics, Newark, CA, USA, Table 1) and a probe specifically targeting *Nkx6.2* mRNA.

### 2.8. Oligosaccharides Analyses in Fecal Samples

To study the profile of carbohydrates in the feces, 100 mg of pellet was mechanically homogenized in a buffer containing 6.7 mM of NaCl and 20 mM of Na_2_HPO_4_. Samples were boiled and centrifuged, and the supernatant was digested with amylase at 20 °C (Table 1). Fecal oligosaccharides were extracted using hydrophilic interaction liquid chromatography-solid phase extraction (HILIC-SPE) technology (Hilicon AB). An aliquot of 250 µL of digested supernatant was diluted (1:1) in acetonitrile and vortex mixed for 30 s before loading it onto the SPE cartridge (50 mg, 1 mL). A mixture of H_2_O–acetonitrile (50:50, 1 mL) and acetonitrile (1 mL) was used for the conditioning and equilibration of the cartridge, respectively. After the loading of the sample, the cartridge was washed with acetonitrile (1 mL) followed by the elution of the sample using 500 µL of H_2_O.

The extracts, diluted in methanol by a factor of 10, were analyzed in negative electrospray ionization mode (ESI) via direct infusion Fourier transform ion cyclotron resonance mass spectrometry (DI-FT-ICR MS), using a Bruker SolariX instrument equipped with a 12-Tesla magnet and an Apollo II ESI electrospray ionization source (ESI) source (Bruker Daltonic GmbH, Bremen, Germany). The instrument was externally calibrated on clusters of arginine (1 mg/mL in methanol/water: 80/20) with calibration errors below 0.1 ppm. The injection flow rate was set to 120 μL/h. One hundred scans were acquired and averaged for each spectrum within the interval from 147.4 to 1000.0 *m*/*z* and with a time domain of 4 megawords (MWs). The voltages of capillary and spray shield were set to 3800 V and −500 V, respectively. The ion accumulation time was set to 200 ms and the time of flight to detector was set to 1 ms. The nebulizer gas flow rate was set at 1 bar and the drying gas flow rate was set to 4 L/min with a temperature of 250 °C.

The acquired spectra were processed using Data Analysis 4.4 software (Bruker Daltonik, GmbH, Bremen, Germany). Peak-picking algorithm was conducted with a signal-to-noise ratio (S/N) of 4 and a minimum intensity threshold of 1.5 × 10^6^ counts. All the peaks were exported as tab-separated asc-files and aligned using an in-house alignment algorithm with the mass tolerance window set to 0.5 ppm. In the generated matrix, *m*/*z* features that occurred in less than 10% of all samples were discarded. Molecular formula assignment was performed following the mass difference network approach [24,25]. The measured intensity levels followed the square root transformation to approximate a normal distribution. The resulting values for each sample were normalized by their total sum. Di-, tri-, and tetra- saccharides were extracted from the generated matrix based on their neutral molecular formulas (C_12_H_22_O_11_ (disaccaride), C_18_H_32_O_16_ (trisaccharide), and C_24_H_42_O_21_ (tetrasaccharide)). The difference between the two groups of interest (Gfi and -mutants and controls) was evaluated with a Student *t*-test.

### 2.9. Quantification and Data Analysis

The total β-cell area and BrdU-labelled cells quantification was performed on pictures from Zeiss Axioimager Z1 using an automatic mosaic acquisition system (Zeiss). Per each sample, at least 5 pancreatic sections were processed for immunohistochemistry. BrdU^+^ cells were manually quantified. For β-cell area quantification, widefield multichannel tile scan image treatment and analysis were performed using Fiji [26] and a homemade semiautomated macro [27]. All values are reported as mean ± SEM of data from at least 4 animals. Data were analyzed using GraphPrism 6 software. Normality was tested using the D’Agostino-Pearson omnibus normality test. Results are considered significant if *p* < 0.0001 (****), *p* < 0.001 (***), *p* < 0.01 (**), and *p* < 0.05 (*).

## 3. Results

### 3.1. Gfi1 Expression in Murine Pancreas

To characterize the *Gfi1* expression pattern in the pancreas, total mRNA was extracted from wild-type mouse pancreas at different developmental and adult (st)ages. *Gfi1* mRNA was readily detected by RT-qPCR at E15.5; its expression levels remained steady until birth (P0, Figure 1A).

During postnatal stages (between P0 and P6) and early adulthood (between 1–3 m old), *Gfi1* mRNA expression levels were found to significantly increase in a stepwise fashion. Thereafter, *Gfi1* remained permanently and steadily expressed at all the stages analyzed, spanning from 3 months to 12 months of age (Figure 1A).

The characterization of *Gfi1* expression pattern in the murine pancreas was hindered by the lack of specific in situ probes or antibodies. Therefore, we generated a reporter mouse line, Gfi1Cre::RosaLac (Figure 1B). In the resulting animals, all cells having once-expressed *Gfi1* were permanently labelled with the *β-galactosidase* reporter gene. Thus, the spatiotemporal expression of *β-galactosidase* driven by the *Gfi1* promoter was assessed by X-gal staining. Importantly, β-galactosidase activity was solely detected within the acinar compartment, while no activity was found in ductal nor endocrine cells (Figure 1C–E). Together, these results demonstrate that *Gfi1* is expressed during most of the animal life; such expression remained exclusively confined to acinar cells.

### 3.2. Loss of Gfi1 Affects the Maturation of Acinar Cells

To assess the role of Gfi1 in the murine pancreas, we generated a conditional knockout mouse line, Pdx1Cre::Gfi1cko (Appendix A). This line was validated by RT-qPCR using primers recognizing the region excised by the Cre recombinase. Importantly, *Gfi1* expression levels from whole organ decreased by 99.8% in the *Gfi1* mutant pancreata compared to controls (Appendix A).

Pdx1Cre::Gfi1cko animals were vital, healthy, and fertile. However, *Gfi1* mutant pancreata appeared slightly hypertrophic upon regular macroscopic examination. Accordingly, the weight of *Gfi1* mutant pancreata, normalized to body weight, was found significantly and progressively augmented during adulthood, with a 13%, 28.5%, and 32.6% increase at 1, 3, and 6 months of age, respectively (Figure 2A). Interestingly, *Gfi1* mutants displayed no difference in pancreas mass at birth, as compared to control counterparts (Figure 2A).

To understand the causes underlying this pancreatic hypertrophy, we assessed the proliferation rate of pancreatic cells. BrdU incorporation analyses revealed rare proliferating cells in 3-month-old control pancreata (Figure 2B). Conversely, a striking number of dividing cells was detected in pancreatic acinar compartment of adult *Gfi1*-deficient mice (Figure 2C). Quantitative analyses confirmed a significant four-fold increase in the number of BrdU^+^ acinar cells in Pdx1Cre::Gfi1cko animals compared to the controls, while no difference was highlighted in the number of endocrine and ductal proliferative cells (Figure 2D). However, macroscopical and immunohistochemical analyses failed did not show any sign of acinar tumors. Similarly, Pdx1Cre::Gfi1cko animals did not present any premature death as compared to their W.T. counterparts.

We pursued our phenotypical analyses of Pdx1Cre::Gfi1cko pancreatic acinar cells by means of immunohistochemical analyses. Interesting results were obtained when focusing on the acinar cell-specific enzyme, amylase. Predictably, amylase immunofluorescence was homogenously detected in all control acinar cells (Figure 3A). However, and most importantly, the majority of acinar cells were found to be negative or weakly positive for amylase immunostaining in Pdx1Cre::Gfi1cko pancreas (Figure 3B). Quantitative PCR-based analysis of *amylase* mRNA confirmed these observations. As expected, *amylase* expression levels increased significantly from E15.5 throughout postnatal and adult life of control mice. Importantly, *amylase* transcript contents of *Gfi1*-deficient pancreatic cells also raised continuously during pre- and post-natal development, but these were found consistently and significantly decreased compared to controls. Strikingly, a massive 78.3% decrease in *amylase* expression levels was measured in 3-month-old *Gfi1* mutants (Figure 3C).

Of note, BrdU and amylase co-immunostaining revealed that amylase-expressing acinar cells were mostly non-proliferative (Appendix A). Specifically, 65.6% of BrdU^+^ cells in the acinar compartment were negative for amylase (Appendix A).

Further immunohistochemical analyses of Pfx1Cre::Gfi1cko pancreata revealed no cytoarchitectural differences between *Gfi1*-deficient and control islets of Langerhans, thus displaying a core of insulin-expressing cells, surrounded by a mantle of glucagon-, somatostatin-, and PP-positive cells (Appendix A).

As expected, rare ghrelin-positive cells were detected in the islets of both control and *Gfi1* knockout adult pancreatic tissue. Remarkably, an impressive number of cells ectopically expressing ghrelin was observed within the acinar compartment of Pdx1Cre::Gfi1cko pancreas (Figure 4A,B). A tridimensional reconstruction of serial sections labeled with ghrelin, insulin, and the ductal marker Dolichos biflorus agglutinin (DBA) allowed us to fully appreciate the striking abundance of *ghrelin*-expressing acinar cells (Appendix A). Interestingly, immunohistochemical characterization of ghrelin-expressing cells in E18.5 pancreatic sections did not highlight any major difference between Pdx1Cre::Gfi1cko samples and control counterparts at this developmental stage. Indeed, these ghrelin^+^ cells were found almost exclusively located within the endocrine compartment of both *Gfi1*-lacking mice and controls. (Appendix A). Intriguingly, in adult *Gfi1* mutant pancreatic sections, ghrelin-immunoreacting cells were found predominantly negative for amylase immunostaining (Figure 4C). Specifically, 76.2% of ghrelin-labelled cells did not co-express amylase, while the remaining 23.8% showed a weak positivity for amylase labeling (Figure 4E). Importantly, a thorough analysis of ghrelin-expressing cells in *Gfi1*-lacking pancreas sections failed to detect any BrdU^+^/ghrelin^+^ cells (Figure 4D). Yet again, we confirmed our immunohistochemical observations by means of RT-qPCR. As previously reported [28], *ghrelin* mRNA was detected in control samples exclusively during embryonic development with a peak at birth followed by a rapid decline (Figure 4F). Conversely and most interestingly, *ghrelin* remained constitutively and increasingly expressed during adulthood in Pdx1Cre::Gfi1cko mice (Figure 4F). Altogether, our data highlight modifications of acinar cells of Pdx1Cre::Gfi1cko mice. Particularly, we found a great number of proliferative cells and ghrelin-expressing cells, which are either weakly positive for, or completely devoid of, amylase.

### 3.3. Gfi1 Mutants Display an Aberrant Overexpression of Nkx6.2

To identify putative Gfi1 target genes, we performed a thorough molecular screen of the main genes involved in pancreatic development and function. Using this approach, we discovered that *Nkx6.2* was abnormally expressed in Pdx1Cre::Gfi1cko pancreas. As previously shown [29], *Nkx6.2* appeared actively expressed during embryogenesis in control pancreas, while being virtually extinguished at 3 months of age. Importantly, *Nkx6.2* was found strongly overexpressed at all stages tested in *Gfi1* knockout animals. Furthermore, *Nkx6.2* mRNA levels did not decline after birth, but rather kept on increasing. Resultantly, an impressive 148-fold increase in *Nkx6.2* expression was observed in adult *Gfi1* conditional knockout animals at 3 months of age, as compared to adult controls (Figure 5A). As no antibodies specifically recognizing Nkx6.2 were commercially available, we opted for RNAscope technology to analyze the expression pattern of *Nkx6.2*. This strategy confirmed that *Nkx6.2* was abundantly expressed in Pdx1Cre::Gfi1cko adult mice, while being undetectable in age-matched controls (Figure 5B,C). Noticeably, RNAscope analyses revealed that *Nkx6.2* mRNA was predominantly expressed in the *Gfi1* mutant pancreatic acinar compartment (Figure 5C). Altogether, our results indicate that Gfi1 is involved in the regulation of pancreatic *Nkx6.2* expression during embryonic development. Consequently, loss of function in Gfi1 is associated with a dramatic overexpression of *Nkx6.2* in adult pancreatic acinar cells.

### 3.4. Pdx1Cre::Gfi1cko Mice Appear Resistant to Diet-Induced Hyperglycemia

*Amylase* expression is strongly downregulated in *Gfi1* mutant pancreatic acinar cells. It is worth reminding that amylase plays a key role in hydrolysis of carbohydrates and, therefore, glucose absorption into the bloodstream. We therefore initially wondered whether blood glucose concentration was altered in Pdx1Cre::Gfi1cko mice. Notably, *Gfi1* mutants were consistently found normoglycemic (Figure 6A). However, their glycemia was systematically slightly lower at all adult ages tested compared to controls (Figure 6A). Intrigued by these observations, we decided to study the effects of *Gfi1* loss of function using experimental models of induced hyperglycemia. Specifically, we fed Pdx1Cre::Gfi1cko animals and controls with a high-fat/high-carbohydrate diet (HFCD) for 20 weeks. Predictably, control animals rapidly developed a chronic and lasting hyperglycemia after 6 weeks (Figure 6B). Strikingly, Pdx1Cre::Gfi1cko animals remained permanently normoglycemic despite long-term HFCD feeding (Figure 6B).

To investigate the capability of *Gfi1* conditional knockout mice to process and absorb carbohydrates, we analyzed fecal oligosaccharides profile of *Gfi1*-lacking mice and controls fed with either chow or HFCD. Specifically, we enzymatically processed fecal complex carbohydrates to oligosaccharides. Subsequently, we extracted and studied the profile of di-, tri-, and tetra-saccharides by liquid chromatography coupled to mass spectrometry. Of note, fecal samples of *Gfi1* mutant mice under chow diet suggested an augmented content in saccharides, when compared to control samples (Figure 6C). Importantly, these experiments revealed a significant increase in fecal saccharides in *Gfi1* knockout mice under HFCD, as compared to the controls (Figure 6D). Altogether, our results provide evidences that the downregulation of *amylase* expression in *Gfi1* mutant acinar cells is associated with an impairment of carbohydrates processing (and thus absorption), leading to an excessive excretion of sugars and lower blood glucose levels. Interestingly, the decreased efficiency of carbohydrates absorption protects *Gfi1* mutant mice against HFCD-induced hyperglycemia.

### 3.5. Loss of Gfi1 Protects Mice against STZ-Induced Diabetes

Phenotypical analyses of Pdx1Cre::Gfi1cko pancreatic samples revealed a significant misexpression of *ghrelin* throughout adulthood. To exert its physiological functions, ghrelin requires a post-translational acylation catalyzed by ghrelin O-acyl transferase (GOAT). Similar to *ghrelin*, *goat* mRNA was mainly detected during intra utero development, its levels dropping postnatally in control pancreatic samples. Conversely, Pdx1Cre::Gfi1cko pancreatic cells failed to extinguish *goat* expression postnatally, which was also found to be consistently and significantly overexpressed in Pdx1Cre::Gfi1cko, as compared to controls (Figure 7A). These results strongly suggest that intrapancreatic ghrelin is successfully acylated and, therefore, activated in *Gfi1* conditional knockout mice. To corroborate this hypothesis, we specifically measured acyl- and deacyl-ghrelin in the pancreas of *Gfi1*-lacking and control mice. Consistent with our gene expression analyses, the concentration of acyl- and deacyl-ghrelin was found strongly and significantly increased in Pdx1Cre::Gfi1cko pancreatic samples, while being hardly detectable in controls (Figure 7B).

Physiologically, ghrelin is a potent orexigenic hormone [30]. However, the assessment of average daily food intake and body weight of adult mice, fed with either chow or HFCD, did not show any significant difference between Pdx1Cre::Gfi1cko and control mice (Appendix A). Importantly, when blood levels of either acyl- or deacyl-ghrelin were assessed by ELISA immunoassay, we failed to detect any significant difference between control and *Gfi1*-deficient samples (Appendix A). Altogether, these data suggest that the ghrelin synthetized by pancreatic acinar cells of *Gfi1* mutant mice is not secreted in the general circulation and, therefore, does not affect feeding behavior.

Beside stimulating food intake, ghrelin was proposed to promote pancreatic β-cells survival in vitro [6,31,32]. We therefore wondered whether a strong increase in intrapancreatic ghrelin, such as the one observed in Pdx1Cre::Gfi1cko mice, might protect β-cells against apoptosis. To verify this hypothesis, we induced β-cell apoptosis by injecting *Gfi1* mutant mice and controls with high doses of STZ. Twenty-four hours after the injection, control islets of Langerhans showed a marked loss of islet cells (Figure 7C,D and Appendix A). In addition, approximately 15% of remaining endocrine cells were found to be apoptotic, as shown by active caspase-3 immunostaining (Appendix A). Conversely, *Gfi1*-lacking endocrine cells did not display any evident sign of degeneration following STZ treatment (Figure 7E). The remaining islet cell numbers were ultimately unaffected, as compared to controls (Appendix A). Furthermore, only 5.5% of *Gfi1* knockout endocrine cells were found positive for active caspase-3 (Figure 7E and Appendix A).

Immunohistochemical analyses of control pancreatic samples performed one week after STZ treatment highlighted the degenerative effect of STZ on β-cells, with an extensive loss of insulin immunoreactivity compared to untreated controls (Figure 7F,G). Consistently, STZ caused a rapid and irrecoverable hyperglycemia in the control mice (Figure 7I). Surprisingly, the total β-cell mass and islets of Langerhans integrity were found largely unaffected upon STZ cytotoxic activity in *Gfi1* knockout pancreata (Figure 7H and Appendix A). Astonishingly, *Gfi1*-deficient mice were found to be resistant to the diabetogenic effects of STZ, displaying an imperturbable normal glycemia for at least 50 days after treatment (Figure 7I). Collectively, our data demonstrated that ghrelin is actively acylated in adult Pdx1Cre::Gfi1cko mouse pancreas. However, this peptide hormone is not released into the bloodstream and does not exert hyperphagic effects. Nevertheless, it displays a surprising protective role against chemically induced β-cell ablation. Indeed, the injection of high doses of STZ caused a negligible loss of the β-cell mass in Pdx1Cre::Gfi1cko mice which remained stably normoglycemic.

## 4. Discussion

Gfi1 is a transcriptional repressor known for its involvement in the development of lymphoid and myeloid cells [33], but also of osteogenic cells [31].

Gfi1 has previously been found to also be expressed in the pancreas, becoming restricted to the tip compartment during the embryonic development [18]. In the present work, we confirm that *Gfi1* is expressed in the mouse pancreatic acinar compartment from in-utero development. This expression is then maintained throughout the entire adult life. To investigate the role of this gene, we used a conditional knockout approach to specifically and constitutively inactivate *Gfi1* in the pancreas. Phenotypical and molecular analyses demonstrated that *Gfi1*-lacking pancreatic acinar cells fail to suppress *Nkx6.2* expression during embryonic development, ectopically express *ghrelin*, and display a strong deficiency in amylase content.

Interestingly, in addition to this altered acinar cell population, we detected that a small subset of *Gfi1* mutant acinar appeared to be quiescent, functional, and negative for ghrelin immunostaining. Physiologically, *Gfi1* is expressed at low levels during embryonic development and in-utero maturation of acinar cells appears mildly affected by *Gfi1* loss of function. We therefore speculate that early-specified acinar cells might succeed to acquire a mature phenotype, while acinar cells arising later during the development show the functional and phenotypical modification we described.

In a previous work, Qu et al. described the phenotypical analysis of a *Gfi1* full knockout mouse model. In these mice, several architectural defects in the acinar compartment were observed, including abnormal amylase immunostaining, as well as reduced expression of several ductal regulatory factors, including Sox9, Hnf1β, and Hnf6 [18]. Gfi1 was therefore suggested to be involved in the development of most distal cells of the pancreatic intercalated duct, i.e., the centroacinar cells. The absence of these cells in *Gfi1* full KO animals would cause the accumulation of digestive zymogens within the acinar cells, leading to phenotypical defects in the acinar unit [18].

In an effort to identify Gfi1 target genes, we analyze the expression of most of the genes involved in mouse pancreatic development and systematically failed to find significant alterations for any ductal marker. While this discrepancy might be due to the different animal models employed, our investigations led us to conclude that Gfi1 plays a role in the maturation of pancreatic acinar cells rather than of centroacinar cells.

Pancreatic amylase is the primary digestive enzyme of starch and glycogen; its activity regulates duodenal glucose absorption, post-prandial increase in glycemia, and undigested carbohydrate excretion. Interestingly, the decrease in *amylase* expression observed in Pdx1Cre::Gfi1cko animals was associated with a significant decrease in basal glycemia. Importantly, blood glucose levels of *Gfi1* mutant mice remained steadily lower compared to controls under chow and, most importantly, under high-fat/high-carbohydrate alimentary regimen. Mass spectrometry analyses of specific fecal sugars confirmed that *amylase* expression defects impairs the ability of Pdx1Cre::Gfi1cko animals to process and assimilate carbohydrates. Accordingly, the levels of oligosaccharides excreted were found tendentially higher in *Gfi1*-lacking mice fed with regular food; this trend became significant when animals were fed with HFCD.

Notably, the continuous misexpression of *Nkx6.2* in multipotent pancreatic progenitor cells has been shown to be sufficient to ablate their differentiation into acinar cells [15], suggesting that this transcription factor hampers the developmental program of acinar cells. We could thus hypothesize that Gfi1 promotes the maturation of pancreatic acinar cells by inhibiting *Nkx6.2* expression during late stages of embryonic development. To investigate this theory, one could cross our Pdx1Cre::Gfi1cko mice with Nkx6.2^-/-^ full knockout animals. While necessitating further work and time, the phenotypical characterization of the resulting Pdx1Cre::Gfi1cko::Nkx6.2^-/-^ mouse line would allow us to determine whether the loss of *Nkx6.2* is sufficient to rescue the developmental program of *Gfi1* mutant acinar cells.

Beside *Nkx6.2*, Pdx1Cre::Gfi1cko acinar cells misexpress another embryonic marker: ghrelin. This hormone is normally transiently produced in the pancreas by a subset of endocrine cells during embryonic development, and its expression rapidly declines after birth. In the absence of *Gfi1*, ghrelin is normally expressed during the embryonic development in the endocrine compartment. However, rather than being silenced, this hormone becomes ectopically expressed in adult acinar cells. It is worth noting that *ghrelin* misexpression has already been described in instances where the inactivation of a transcription factor was preventing the specification of a particular pancreatic cell lineage. For instance, targeted disruption of *Nkx2.2* in mice results in a complete loss of β-cells and reduced numbers of α- and PP-cells, which are then replaced by ghrelin^+^ cells [32]. Similarly, in *Pax4* (Paired *box* gene 4) full knockout animals, endocrine precursors fail to differentiate into insulin-producing cells and, rather, become glucagon/ghrelin co-expressing cells [34]. Here, we show that the ablation of *Gfi1* is associated with an increase in *ghrelin*-expressing cells at the expense of amylase-producing acinar cells.

Terminal differentiation of pancreatic cells is classically coupled with permanent exit from the cell cycle. Phenotypical analyses of Pdx1Cre::Gfi1cko pancreata revealed a significant increase in the number of dividing acinar cells, eventually leading to a pancreatic hypertrophy. Interestingly, ghrelin has been shown to increase pancreatic cell proliferation and accelerate pancreatic weight gain in a rat model of cerulein-induced acute pancreatitis [35]. Our phenotypical analyses of Pdx1Cre::Gfi1cko pancreata therefore indicate that ghrelin expressed in adult *Gfi1* mutant pancreatic cells might exert a paracrine/autocrine hyperproliferative effect.

In vitro, ghrelin has been demonstrated to protect β-cells against apoptosis induced by serum starvation, treatment with cytokines, or exposure to dexamethasone [36,37]. Furthermore, ghrelin treatment prevented diabetes development in STZ-treated neonatal rats [38]. To assess whether acinar-derived ghrelin might protect β-cells against apoptosis, we injected *Gfi1* mutant mice with high doses of STZ. Impressively, STZ exerted a negligible cytotoxic effect on β cells of *Gfi1* mutant mice, but failed to affect their glycemia or β-cell mass. As *Gfi1* is not expressed in pancreatic endocrine cells at any time of embryonic development or adulthood, these results strongly indicate that the sole inactivation of *Gfi1* and the consequent misexpression of *ghrelin* in the acinar cells are sufficient to protect β-cells against chemically induced diabetes.

Several aspects of the role of ghrelin misexpression in the glucose metabolism and β-cell survival of Pdx1Cre::Gficko mice still remain to be elucidated. Interestingly, ghrelin has also been proposed to modulate insulin secretion, to prevent insulin resistance and to contribute to glucose handling [39]. It would thus be of interest to determine whether an increase in the intrapancreatic levels of ghrelin observed in *Gfi1*-deficient animals could also modulate insulin secretion and/or sensitivity.

However, the role of ghrelin on insulin release remains highly controversial. Indeed, this hormone has been reported to exert either a stimulatory or an inhibitory effect on β-cell secretion [40]. Moreover, while some reports indicate that ghrelin acts directly on β-cells by increasing K_ATP_ current [41], other studies suggest that ghrelin regulates indirectly β-cell activity by an indirect mechanism, whereby ghrelin would act on δ-cells to stimulate somatostatin release, which in turn would inhibit insulin release from β-cells [42,43]. Finally, while Dezaki et al. suggested that intrapancreatic ghrelin acts locally on the regulation of insulin release [44], other reports indicate that islet ghrelin is dispensable for insulin secretion. To fully dissect the effects of acinar ghrelin on the islets of Langerhans, it would be interesting to generate a mouse model in which this hormone would be conditionally misexpressed in the adult pancreatic exocrine compartment. As acinar ghrelin is seemingly not secreted in the general circulation, this model would offer the possibility to evaluate the paracrine effects of ghrelin misexpression on pancreatic cells, without interfering with other ghrelin target organs. The evaluation of glucose-stimulated insulin secretion in such animals would also shed a new light on the role of intrapancreatic ghrelin on β-cell activity. Alternatively, it has been recently shown that ghrelin activity can be antagonized by liver-expressed antimicrobial peptide 2 (LEAP-2) [45]. Investigation and genetic manipulation of LEAP-2 specifically in the pancreas could represent an additional tool to thoroughly dissect the activity of pancreatic ghrelin in *Gfi1* mutant mice.

In summary, our results demonstrate that, in the absence of *Gfi1*, pancreatic acinar cells fail to become terminally differentiated, fully functional, and postmitotic cells. The resulting decrease in amylase synthesis affects intestinal carbohydrates processing and glucose absorption, and Pdx1Cre::Gfi1cko mice remained consistently normoglycemic, even when challenged with long-term HFCD. Acinar cell also ectopically express ghrelin in *Gfi1*-deficient pancreata. This hormone exerts a paracrine action on the pancreatic endocrine cells, protecting *Gfi1* mutant mice against STZ-induced diabetes. Altogether these data show that targeting Gfi1 signaling pathway results in the simultaneously inhibition of glucose absorption under a diabetogenic diet and exerts a potent anti-apoptotic effect on β-cells. In other words, inhibiting Gfi1 could be extremely beneficial for both type 1 diabetes mellitus and type 2 diabetes mellitus as it prevents diet-induced hyperglycemia while sustaining the survival of β cells.

## Figures and Tables

**Figure 1 cells-10-02805-f001:**
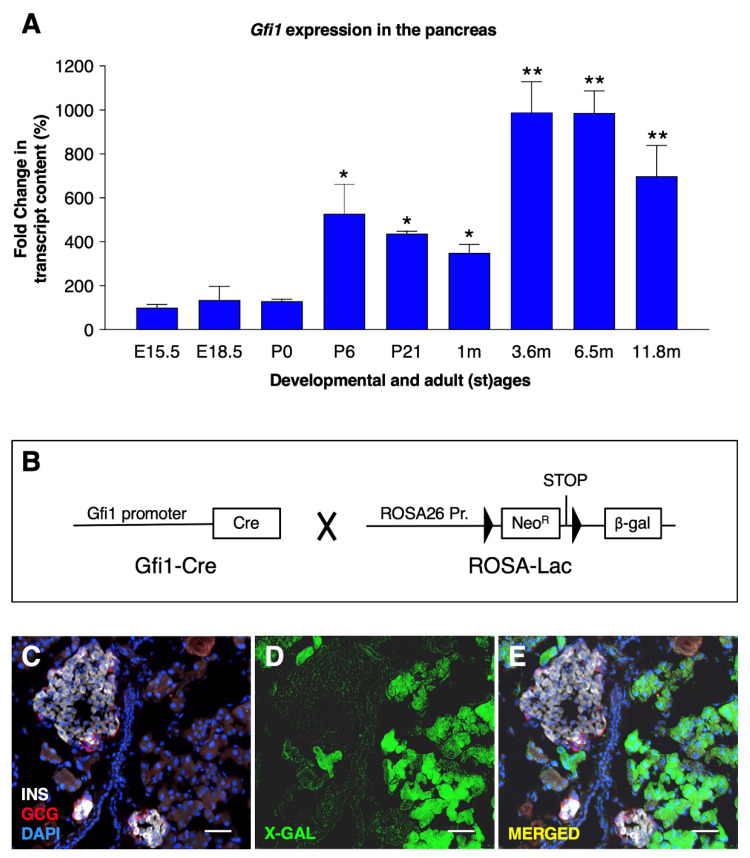
*Gfi1* expression in the mouse pancreas. The expression levels of *Gfi1* were assessed by RT-qPCR using RNAs from whole pancreata from control mice (*n* = 6) at different developmental and adult (st) ages (* *p* < 0.05, ** *p* < 0.01). (**A**). Antibodies specifically recognizing Gfi1 are not commercially available. The *Gfi1* expression pattern was therefore determined using a transgenic mouse line expressing a reporter gene under the control of *Gfi1* promoter (**B**). Specifically, Gfi1-Cre animals (the *Cre recombinase* being knocked-in into the *Gfi1* locus) were crossed with the Rosa26-β-gal mice (harboring a transgene composed of the ubiquitous *Rosa26* promoter upstream of a LoxP-STOP-LoxP cassette followed by the *β-galactosidase cDNA)*. An X-gal staining was performed on 4-month-old Gfi1Cre::RosaLac animals (the blue channel was converted in green to increase the image readability) to reveal β-galactosidase activity. Objective magnification: 20×. Scale bar: 50 μm (**C**–**E**).

**Figure 2 cells-10-02805-f002:**
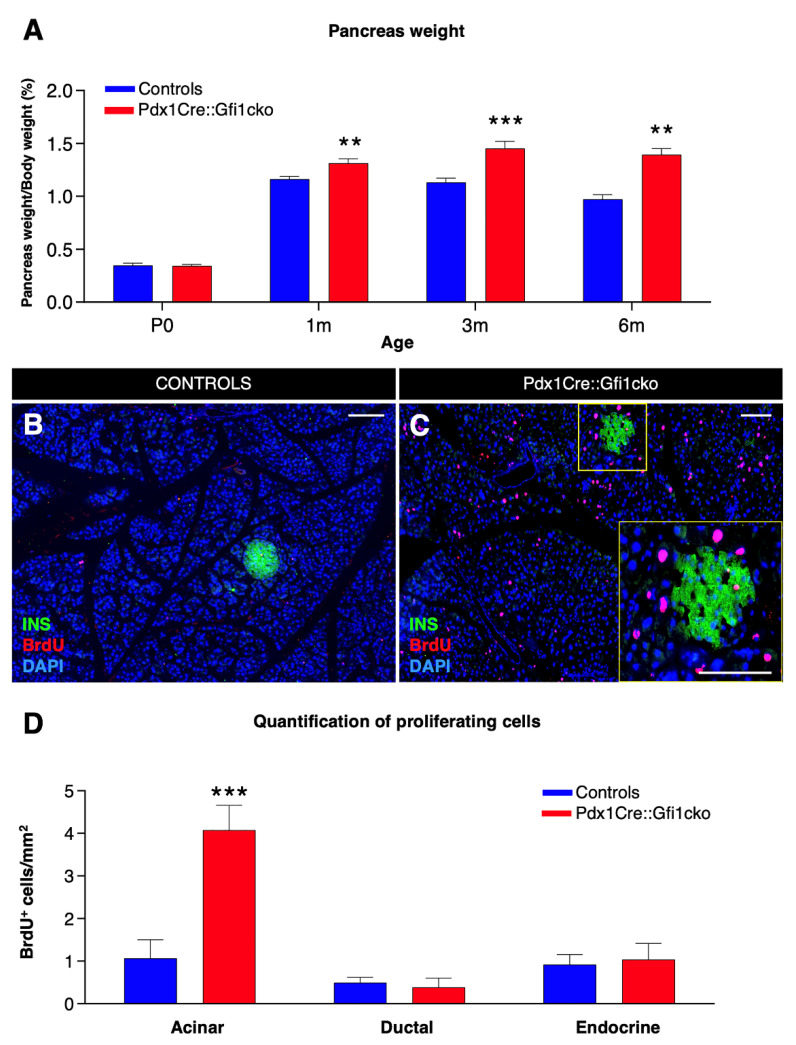
Loss of *Gfi1* is associated with a dramatic increase in proliferation rate within the acinar compartment. Measurement of pancreas weight normalized to animal body weight at birth and 3 adult ages. (*n* = 8 animals per genotype per (st) age) (** *p* < 0.01, *** *p* < 0.001) (**A**). Adult Pdx1Cre::Gfi1cko and controls were provided with BrdU for 5 days. Subsequently, pancreatic sections were analyzed by immunofluorescence. Objective magnification: 20×. Scale bar: 100 μm (**B**,**C**). Quantitative analyses of BrdU-labelled cells in different pancreatic compartments (*n* = 4 animals per genotype) (*** *p* < 0.001) (**D**).

**Figure 3 cells-10-02805-f003:**
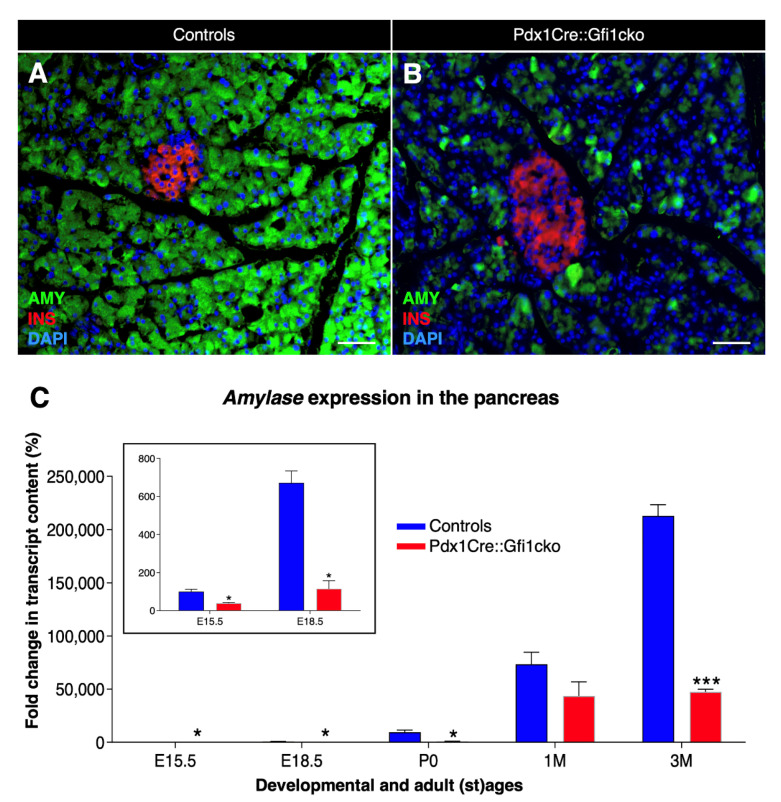
Important decrease in amylase expression in Pdx1Cre::Gfi1cko animals. Immunohistochemical analyses of Pdx1Cre::Gfi1cko and control mice using antibodies recognizing insulin (red) amylase (green) and DAPI (blue). Objective magnification: 20×. Scale bar: 50 μm (**A**,**B**). *Amylase* expression levels were assessed by RT-qPCR at different developmental and adult ages. Insert graph shows *amylase* expression at E15.5 and E18.5 of both *Gfi1*-conditional knockouts and controls. (*n* = 5 animals per genotype per (st) age) (* *p* < 0.05, *** *p* < 0.001) (**C**).

**Figure 4 cells-10-02805-f004:**
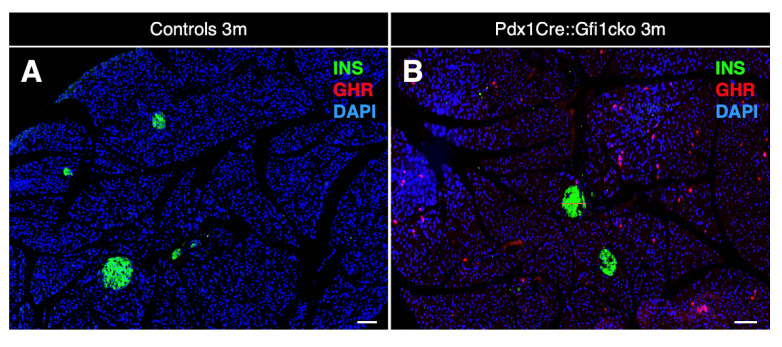
Ghrelin is misexpressed in Pdx1Cre::Gfi1cko adult pancreas. Immunohistochemical assessment of ghrelin-labeled cells in 3-month-old Pdx1Cre::Gfi1cko adult acinar cells. Objective magnification: 10×. Scale bar: 100 μm (**A**,**B**). Immunohistochemical analyses of amylase- and ghrelin-expressing cells in *Gfi1*-deficient pancreatic tissue. Objective magnification: 20×. Scale bar: 20 μm (**C**). Quantification of amylase and ghrelin co-immunoexpressing cells in *Gfi1*-deficient pancreatic acinar compartment (*n* = 253 ghrelin^+^ cells) (**E**). Phenotypical evaluations of ghrelin and BrdU labelled cells by immunofluorescence in Pdx1Cre::Gficko adult pancreas. Objective magnification: 20×. Scale bar: 50 μm (**D**). Quantitative evaluation of pancreatic *ghrelin* expression by RT-qPCR at different developmental and adult (st)ages (*n* = 5 animals per genotype per (st) age) (*** *p* < 0.001) (**F**).

**Figure 5 cells-10-02805-f005:**
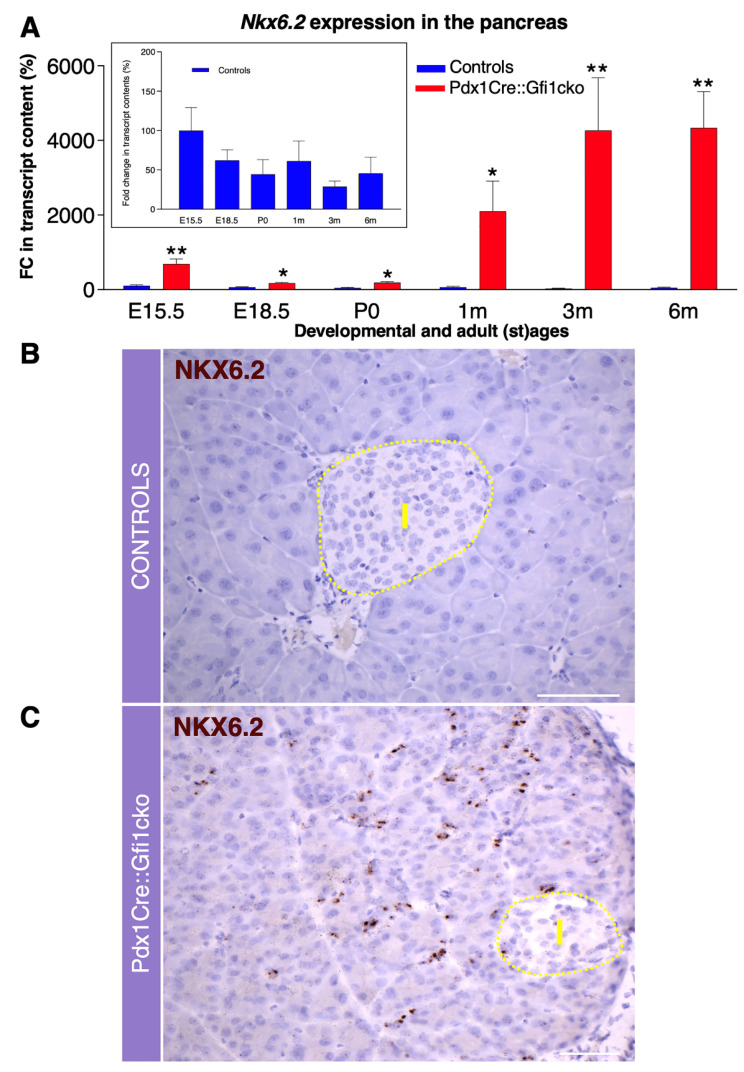
Gfi1 is crucially involved in the regulation of *Nkx6.2* expression in mouse pancreas. Expression levels of *Nkx6.2* measured by RT-qPCR during the embryonic development and adulthood in the pancreas of Pdx1Cre::Gfi1cko animals and age-matched controls (*n* = 5 animals per genotype per (st) age). Insert graph shows *Nkx6.2* expression exclusively in control pancreatic samples. (* *p* < 0.05, ** *p* < 0.01) (**A**). *Nkx6.2* expression pattern was assessed by RNA scope in adult *Gfi1* loss of function and control pancreatic sections. Yellow dash line indicates the islet compartment. Objective magnification: 25×. Scale bar: 50 μm (**B**,**C**).

**Figure 6 cells-10-02805-f006:**
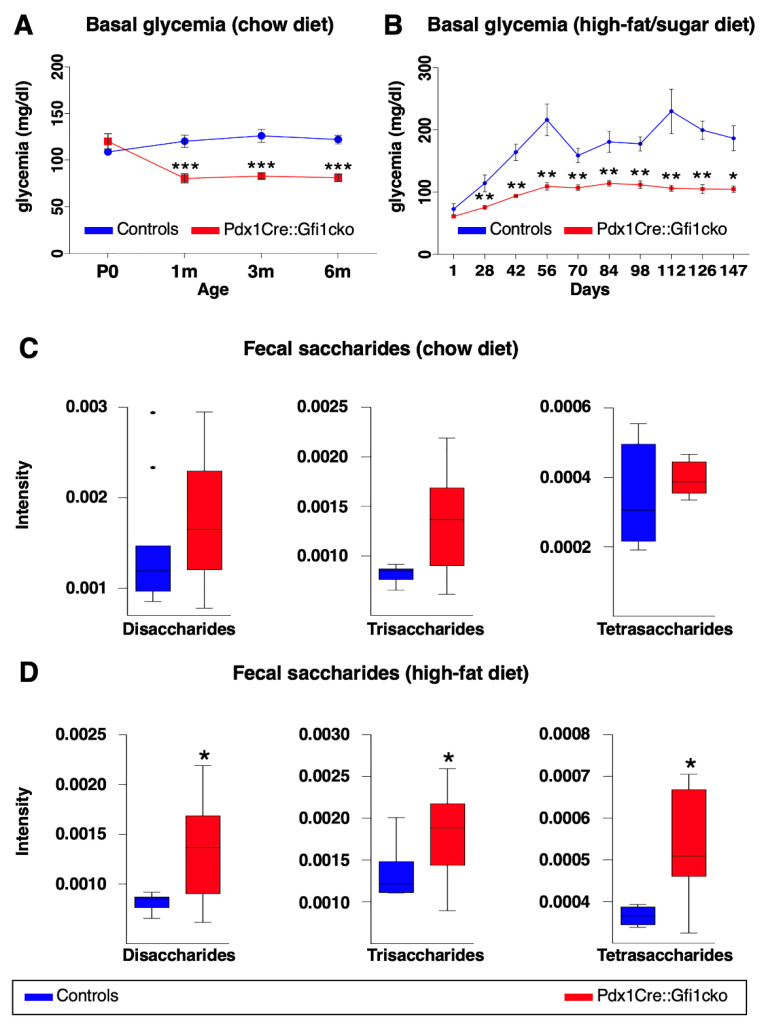
Pdx1Cre::Gfi1cko mice are resistant to high-fat/high-sugar diet-induced hyperglycemia. Basal glycemia was assessed in control and *Gfi1*-deficient mice (*n* = 8) at different ages. (*** *p* < 0.001) (**A**). Control and *Gfi1* mutant mice were fed with HFCD (*n* = 7 animals per genotype) and their glycemia was monitored for 5 months. (* *p* < 0.05, ** *p* < 0.01) (**B**). Semiquantitative assessment of fecal saccharides by mass spectrometry was performed using samples of Pdx1Cre::Gfi1cko mice and controls fed with chow diet (**C**) and HFCD (**D**). (chow diet, *n* = 11 for control and 16 for Pdx1Cre::Gfi1cko samples; HFCG, *n* = 5 for control and 12 for Pdx1Cre::Gfi1cko samples) (* *p* < 0.05).

**Figure 7 cells-10-02805-f007:**
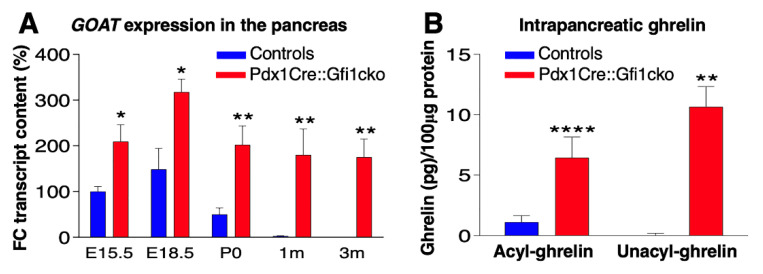
Pdx1Cre::Gfi1cko mice are protected against chemically-induced diabetes. The expression levels of *goat* were assessed by RT-qPCR in *Gfi1* mutant and control pancreatic samples during embryonic development and postnatally (*n* = 5 animals per genotype per (st) age) (* *p* < 0.05, ** *p* < 0.01) (**A**). ELISA immunodetection of acyl- and deacyl-ghrelin was performed on protein lysates from whole pancreatic tissue extracted from control and *Gfi1* mutant mice. (*n* = 6 animals per genotype) (** *p* < 0.01, **** *p* < 0.0001) (**B**). Immunohistochemical evaluation of pancreatic islets of Langerhans targeting active caspase-3 (in red) of control and *Gfi1* mutant mice 24 h after treatment with or without high doses of STZ. Objective magnification: 20×. Scale bar: 50 μm (**C**–**E**). Immunohistochemical assessment of β-cell mass was performed on sections isolated from control and Pdx1Cre::Gficko mice following 7 days of STZ treatment (**F**,**H**) and non-treated controls (**G**). Objective magnification: 10×. Scale bar: 500 μm. Basal glycemia of STZ-treated control and *Gfi1*-deficient mice for 7 weeks. (*n* = 8 animals per genotype) (** *p* < 0.01, *** *p* < 0.001, **** *p* < 0.0001) (**I**).

**Table 1 cells-10-02805-t001:** Key reagents and resources.

Reagent or Resources	Source	Identifier
**Antibodies**		
Guinea pig polyclonal anti-insulin	Dako, Hovedstaden, Denmark	Ref: A0564
Rat monoclonal anti-BrdU	Abcam, Cambridge, UK	Ref: ab6326
Rabbit polyclonal anti-pancreatic amylase	ThermoFisher, Waltham, MA, USA	Ref: PA5-25330
Mouse monoclonal anti-ghrelin	Merck, Readington Township, NJ, USA	Ref: MAB10404
Rabbit polyclonal active caspase-3	R&D Systems, Minneapolis, MN, USA	Ref: AF835
Rabbit monoclonal anti-proglucagon	Cell Signaling, Danvers, MA, USA	Ref: 8233
Rabbit polyclonal anti-somatostatin	Dako, Hovedstaden, Denmark	Ref: A0566
Rabbit polyclonal anti-pancreatic polypeptide	Millipore, Burlington, MA, USA	Ref: AB939
Goat anti-guinea pig IgG (H+L) Alexa Fluor 488 conjugated	ThermoFisher, Waltham, MA, USA	Ref: A-11073
Goat anti-guinea pig IgG (H+L) Alexa Fluor 594 conjugated	ThermoFisher, Waltham, MA, USA	Ref: A-11076
Goat anti-rat IgG (H+L) Alexa Fluor 594 conjugated	ThermoFisher, Waltham, MA, USA	Ref: A-11007
Donkey anti-Rabbit IgG (H+L) Alexa Fluor 488 conjugated	ThermoFisher, Waltham, MA, USA	Ref: A-21206
Goat anti-Mouse IgG (H+L) Alexa Fluor 594 conjugated	ThermoFisher, Waltham, MA, USA	Ref: A-11005
Donkey anti-Rabbit IgG (H+L) Alexa Fluor 594 conjugated	ThermoFisher, Waltham, MA, USA	Ref: A-21207
**Experimental Models: Organisms/Strains**		
Pdx1-Cre	[21]	N/A
Gfi1cko	[22]	N/A
Gfi1-Cre	[19]	N/A
ROSA26-lox-Stop-lox-β-Gal	[20]	N/A
**Chemicals, Peptides, and Recombinant Proteins**		
d-(+)-glucose	Sigma-Aldrich, St. Louis, MO, USA	Ref: G7528-1KG
Insulin	Novo Nordisk, Bagsvaerd Denmark	N/A
Streptozocin	Sigma-Aldrich, St. Louis, MO, USA	Ref: S0130
RNAlater	Invitrogen, Waltham, MA, USA	Ref: AM7021
Collagenase	Sigma-Aldrich, St. Louis, MO, USA	Ref: C7657
Histopaque-1077	Sigma-Aldrich, St. Louis, MO, USA	Ref: 10771
Histopaque-1119	Sigma-Aldrich, St. Louis, MO, USA	Ref: 11191
Antigenfix	Microm Microtech, Brignais, France	Ref: F/P0016
Antigen Unmasking Solution	CliniSciences, Nanterre, France	Ref: H-3300
Tissue freezing medium	Leica, Wetzlar, Germany	Ref: 14020108926
Amylase from porcine pancreas	Sigma-Aldrich, St. Louis, MO, USA	Ref: A6255
**Critical Commercial Assays**		
RNeasy Mini Kit	Qiagen, Hilden, Grmany	Ref: 74104
Unacylated Ghrelin (mouse, rat) Express Enzyme Immunoassay kit	Bertin Bioreagent, Frankfurt am Main Germany	Ref: A05118
Acylated Ghrelin (mouse, rat) Express Enzyme Immunoassay kit	Bertin Bioreagent, Frankfurt am Main Germany	Ref: A05117
Pierce™ BCA Protein Assay Kit	ThermoFisher, Waltham, MA, USA	Ref: 23225
RNAscope 2.5 HD Reagent Kit-BROWN	Advanced Cell Diagnostics, Newark, CA, USA	Ref: 322370
**Software and Algorithms**		
Prism software (Graphpad)	GraphPad Software, Inc. 7	N/A
FIJI	imageJ-win64	N/A
ZEISS Axiovision	Carl Zeiss AG Axiovision Rel. 4.8	N/A
**Others**		
ONETOUCH Vita glucometer	Life Scan, Milpitas, CA, USA	N/A

## Data Availability

The data presented in this study are available on request from the corresponding author. The data are not publicly available due to their lack of a dedicated repository.

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
