# Peer review of "Gfi1 Loss Protects against Two Models of Induced Diabetes"

_cells, 2021, doi:10.3390/cells10112805_

Round 1
Reviewer 1 Report
In this study, Tiziana and her coauthors first generated animals deficient for Gfi1 specifically in the pancreas. They observed that Gfi mutant mice did not show any obvious deleterious phenotype. Importantly, they demonstrated that Gfi1-deficient mice were found resistant to diet-induced hyperglycemia, remaining steadily normoglycemic following STZ treatment. Overall, it is an interesting study, and the novelty and strengths of this manuscript are high. However, there are some critical points that should be addressed:
- Many abbreviations in this manuscript should be defined. For instance, what is Gfi1?
- The authors should describe what’s the effect of high proliferation rate within the adult Gfi1 mutant acinar compartment? Does it lead to acinar tumorigenesis?
- Images from some IHC experiments need to add scale bars.
- It would be interesting to test whether Ghrelin can directly protect against chemically-induced diabetes.
Reviewer 2 Report
I really enjoyed reading this manuscript entitled “Gfi1 loss protects against two models of induced diabetes”, which suggests that Gfi1 is a transcription factor that plays a key role in the differentiation and full maturation of acinar cells. I do, however, have some suggestions for improvement. Please find below my comments to the authors.
General comments
1) Page 2, line 60: the abbreviations for “type 1 diabetes mellitus” appear only twice in the manuscript. Thus, there is no need to abbreviate “type 1 diabetes mellitus”.
2) Page 2, line 61: “(…) pancreatic -cells loss [7].” – the symbol “β” for “β-cells” is missing.
3) Page 2, line 62: the abbreviations for “type 2 diabetes mellitus” appear only twice in the manuscript. Thus, there is no need to abbreviate “type 2 diabetes mellitus”.
4) Page 2, line 71: the abbreviations for “multipotent pancreatic progenitor cells” appear only twice in the manuscript. Thus, there is no need to abbreviate “multipotent pancreatic progenitor cells”.
5) Page 2, line 116: it is not clear at which age mice were treated with streptozotocin. Please include this information in the manuscript.
6) Page 3, line 121: please include a reference that describes the protocol for islet isolation by collagenase digestion used in the present work.
7) Page 10, lines 292-294: “Indeed, in these ghrelin+ cells (…) and controls.” – The preposition “in” does not make much sense in this sentence. Please check whether this observation is correct and, if so, correct the sentence accordingly.
8) Page 10, lines 302-304: “Conversely and most interestingly (…) in Pdx1Cre::Gfi1cko mice (Figure 4F).” – Did the authors measure ghrelin expression in mice older than 3-month-old to check whether ghrelin expression would decrease at some point just like ghrelin expression in control mice did? As the authors already have the samples from different (st)ages (shown in Figure 1), this should be a straightforward experiment. Moreover, it might help to explain why Gfi1-deficient mice were found to be resistant to STZ treatment and normoglycemic for at least 50 days after treatment.
9) Page 12, lines 326-327: “Furthermore, Nkx6.2 mRNA levels (…) but rather kept on increasing.” – I would like to know whether the authors measured Nkx6.2 expression in mice older than 3-month-old to check whether Nkx6.2 expression would decrease at some point.
10) References 28 to 37 are missing: this is likely a mistake during the insertion of the bibliography in the final version of the manuscript. Please correct it accordingly.
Figure legends
11) It may be a matter of style, but I believe the figure legends should only contain the information necessary to understand the figures to which such captions refer to. Thus, I suggest the authors to avoid describing the results again in the figure legends once they have been already described in the appropriate section of the manuscript. Moreover, some figure legends are too long due to the description of results.
12) Figure 1: please indicate the age of the animals used in the experiments depicted in Figure 1C.
13) Figure 5: please indicate what the area marked in yellow means.
Figures
14) All figures involving bar graphs: while the fonts used for the axis titles are clear, the size of the fonts used for most numbers in y- and x-axis are too small in the printed version. I suggest the authors to change them to increase readability.
15) Figure 3: the x-axis of the inset graph (Fig. 4C) is not clear in the printed version. Please change it to increase readability. Of note, I believe the inset should be mentioned in the figure legend.
16) Figure 5: due to the huge Nkx6.2 overexpression observed in Gfi1-deficient mice, it is difficult to appreciate the increase in Nkx6.2 during embryogenesis followed by its disappearance at 3-months of age. Therefore, I suggest the authors to introduce an inset graph showing Nkx6.2 expression only in control mice.
17) Figure 7: in the y-axis title of Fig. 7B, the word “pg” is underlined in red. Please correct it accordingly.
Round 2
Reviewer 1 Report
The authors have addressed my comments, but please provide the animal IACUC permission number for this study before publishing.